# Synthesis, Chemical and Biomedical Aspects of the Use of Sulfated Chitosan

**DOI:** 10.3390/polym14163431

**Published:** 2022-08-22

**Authors:** I. N. Bolshakov, L. M. Gornostaev, O. I. Fominykh, A. V. Svetlakov

**Affiliations:** 1Department of Operative Surgery and Topographic Anatomy, FSBE Higher Education Prof. V.F. Voyno-Yasenetsky Krasnoyarsk State Medical University, Krasnoyarsk 660022, Russia; 2Department of Biology, Chemistry and Ecology, Krasnoyarsk State Pedagogical University Named after V.P. Astafiev, Krasnoyarsk 660049, Russia; 3AlfaChem Limited Liability Company, Krasnoyarsk 660135, Russia

**Keywords:** organic synthesis, O,N-(2-sulfoethyl)chitosan, sulfated polyelectrolyte complexes, transport system, experimental atherosclerosis, para-adventitial implantation, vascular cholesterol extraction

## Abstract

This work is devoted to the chemical synthesis of sulfated chitosan and its experimental verification in an animal model of early atherosclerosis. The method of chitosan quaternization with sulfate-containing ingredients resulted in a product with a high content of sulfate groups. Implantation of this product into the fascial-muscular sheath of the main limb artery along the leg and thigh in rabbits led to the extraction of cholesterol from the subintimal region. Simplified methods for the chemical synthesis of quaternized sulfated chitosan and the use of these products in a model of experimental atherosclerosis made it possible to perform a comparative morphological analysis of the vascular walls of the experimental and control limbs under conditions of a long-term high-cholesterol diet. The sulfated chitosan samples after implantation were shown to change the morphological pattern of the intimal and middle membranes of the experimental limb artery. The implantation led to the degradation of soft plaques within 30 days after surgical intervention, which significantly increased collateral blood flow. The implantation of sulfated chitosan into the local area of the atherosclerotic lesions in the artery can regulate the cholesterol content in the vascular wall and destroy soft plaques in the subintimal region.

## 1. Introduction

### 1.1. Methods for the Synthesis of Sulfated Chitosans

It is known that water-soluble derivatives of chitosan exhibit various types of biological activity [1]. The prospect of using sulfated chitosan in the correction of atherosclerotic lesions in the great vessels has stimulated researchers to develop new and simple methods for the synthesis of sulfated polysaccharides [2,3]. Water-soluble O,N-(2-sulfoethyl)chitosans (SECs) are of interest and have not been previously studied in detail. It is known that SEC derivatives, which are synthesized by the intermolecular cross-linking of chitosan molecules with glutaric dialdehyde, are suitable for the separation and concentration of metal ions (silver, copper, zinc, nickel, cobalt, cadmium, lead, manganese, calcium, strontium, magnesium, and barium) [4].

The authors of [5] proposed a method for SEC synthesis using pre-exposure of a water suspension of chitosan to ultrasound (3 min, 300 W), followed by treatment with sodium dichloroethylene sulfonate in a nitrogen atmosphere for 3 h at 80 °C. The yield of sulfoethylated chitosan depended on the molecular weight of the starting molecule. In previously published works [6,7], the yield of O,N-(2-sulfoethyl)chitosan was rather low in terms of 100% sulfation.

An interesting method for the synthesis of sulfoethylated chitosans is the interaction of chitosan with 1,3-propane sulfonate in an ionic medium, which consists of glycine and hydrochloric acid [8]. The degree of sulfation was 0.76. The resulting product had pronounced antimicrobial activity, which stimulated research in this direction. A convenient method for obtaining sulfated chitosans was proposed in the works of Pestov A.V. [8,9,10,11]. Sonat chitosan (250 kDa) was reacted with sodium 2-bromoethyl sulfonate in water for gel formation. Selective sulfoethylation only at the amino group took place in the gel in the presence of a small amount of hydrochloric acid at 70 °C. The disadvantage of this method is the hours-long extraction of the product in a Soxhlet apparatus. We have significantly improved the method for the synthesis of sulfoethylated chitosan proposed in [8,9,10,11]. Chitosans of different molecular weights were converted to a gel by treatment with hydrochloric acid and sodium bromoethylene sulfonate.

It turned out that high-molecular-weight chitosan (Bioprogress, Schelkovo, Russia, 500 kDa) did not form a gel and practically did not react with sodium bromoethylene sulfonate. Sonat chitosan (250 kDa) and LLC NATEK chitosan (100 kDa) interacted with sodium bromoethylene sulfonate, with better results obtained in the case of Sonat chitosan.

### 1.2. Mechanisms of Formation of Vascular Atherosclerosis and Signs of Cholesterol Extraction

Lower extremity peripheral arterial occlusive disease (OPAD) is an important and very common manifestation of systemic atherosclerosis. The disease leads to the significant restriction of patients’ ability to move and a decrease in their quality of life and is associated with a high risk of cardiovascular morbidity and mortality [12,13]. The prevalence of OPAD, mainly in the lower extremities, is steadily increasing [14,15]. At the same time, the development of critical lower limb ischemia (CLLI) indicates the complete decompensation of blood circulation and occurs with a frequency of 400–1000 cases per 1 million population per year, or in 15–20% of patients with OPAD [16,17]. The expected mortality of patients with CLLI increases from 25% during the first year of the syndrome to 60–70% over the next three–five years. Indications for high, that is, above the level of the knee joint, amputation reaches 52–95% within three years and is accompanied by total mortality of 10–40% to 71% over the next two–three years [18].

The most important vascular damage factor is a high level of low-density lipoproteins (LDL) [19,20,21]. The accumulation of a mass of oxidized forms of LDL in the subendothelial space enhances the dysfunction of the vascular endothelium, increases the adhesiveness of endothelial cells, and forms lipid and cellular leaks. Monocytes of the vascular bed migrate into the subendothelial zone, and the uptake of lipid mass by cells increases many times [22]. An atherogenic inflammatory reaction is characterized by the deposition of a free mass of cholesterol crystals in the subintimal zone of the vessel and the proliferation of smooth muscle cells in the middle layer of the vessel with a disruption of their orientation and migration to the intimal zone. The expression of residual blood cell receptors and multicellular destruction in all layers of the vessel indicates that the inflammatory reaction goes beyond the adventitia [23]. To solve the problem of local control of the inflammatory reaction in the layers of atherosclerotic vessels, the authors decided to implant artificial chitosan-based polysaccharide directly into the damaged area. In this case, improving the vascularization of the ischemic zone along with the destruction of atherosclerotic plaques in the subendothelial zone of the vessels is a very important end goal. Methods of indirect vascularization and gene therapy known in clinical practice [24,25,26] demonstrate undifferentiated implantation in the ischemic zone of the extremities. The introduction of polymeric water-soluble polysaccharide structures into the perivascular space of the main arteries over a large length of the vessel poses the problem of targeted implantation [27]. The adventitia layer of the vessel is functionally associated with the middle and intimal layers. The improved adventitia of the vessel can regulate the proliferation of smooth muscle cells in the middle and endothelial layers of the vessel, which is important in the treatment of atherogenic inflammation [28].

The implantation of nanosized polysaccharides in the perivascular space ensures easy penetration of the product into the intimal and middle layers of the vessel and the direct capture and strong binding of cholesterol crystals and lipids of foam cells. The need for minimally invasive relocation of the center of atherogenic inflammation from the subintimal to the adventitial space requires scientific confirmation. The experimental use of noncovalent and covalent copolymers that contain chitosan and a sulfated ingredient, e.g., heparin, confirms this concept [29,30,31]. The destruction of atherogenic plaques, especially at an early stage of inflammation, is largely based on the known fact of the binding of positively charged chitosan molecules to negatively charged fatty acid residues, thus confirming the lipophilic nature of the polymer [32]. The lack of toxicity of the polymer during its long-term implantation in tissues may be an additional positive effect of the action of chitosan [33,34]. A decrease in the inflammatory response in the presence of chitosan may be due to the draining effect of the polymer, which explains the morphological preservation of tissues at the site of the polymer dislocation and in its vicinity against the background of a systemic destructive process [35,36,37].

The introduction of polycationic biopolymers in the vicinity of the intimal membrane of the vascular wall, which is affected by cholesterol invasion, suggests the implementation of several interrelated mechanisms of local decholesterolization. The high penetrating ability of nanosized chitosan polyelectrolyte complexes through vessel membranes enables the delivery of compounds with high affinity to cholesterol. The use of sulfated chitosans for parenteral correction of lipid metabolism in the vascular wall in atherosclerotic lesions is a novel direction. The main task is to deliver the cholesterol capture system to macrophages and transport the water-soluble system outside the cells and the subintimal zone.

It is known that sulfation of the amino group of chitosan leads to the formation of an anionic heparin-like water-soluble polysaccharide [38,39]. This product is used for vascularization and tissue regeneration [40]. Implants made of nanoparticles based on 2-N,6-O-sulfated chitosan improve angiogenesis and form tissue, e.g., bone, with a rich network of blood vessels [41]. These regeneration mechanisms are related to the mucoadhesive properties of the polymer and cell permeability, which increases the efficiency of the local delivery of target ingredients. The positive charge ensures the adhesion of chitosan materials to negative substances due to electrostatic interactions. Moreover, the interaction of chitosan with the negatively charged cell wall disrupts the alignment of cell membrane phospholipids and promotes penetration into cells [42,43]. Polyelectrolyte complex hydrogels that consist of chitosan and sulfated glycosaminoglycans (GAGs) have attracted considerable attention because of not only their compatibility and biological activity but also, in our case, the hypocholesterolemic functions of the complexes [44].

Direct interactions of chitosan with sulfated polysaccharides are accompanied by the self-assembly of polyelectrolyte complexes (PECs) in the form of biocompatible hydrogels, which are sufficiently stable in tissues [45,46]. It is known that the physical combination of chitosan with sulfated polysaccharides leads to the twisting of the linear polymer due to electrostatic interactions, hydrogen bonds, and the formation of a core of nanoparticles [47,48,49,50]. This core can include hydrophobic compounds for their transfer.

The data in [50] indicate that the particle size of 1000 kDa chitosan is in the range of 48–52 nm depending on the configuration of the molecular chain. It should be noted that the molecular weight and size of chitosan nanoparticles significantly depend on the medium pH value. A change in pH from 1.55 to 3.5 leads to a decrease in the size of nanoparticles from 200–220 nm to 48–52 nm because of the pH-dependent change in the molecular conformation [50]. The particle size also decreases during chemical degradation. As shown in [51], chitosan with an initial molecular weight of 800 kDa and degree of deacetylation of 96% degraded to molecules with weights in the range from 50 to 100 kDa and nanoparticle sizes in the range from 20 to 120 nm [52]. It should be noted that the efficiency of chitosan molecules depends on the charge state of the biopolymer. In acidic media, chitosan transforms into a water-soluble cationic form, which promotes the electrostatic binding to anionic components of the medium [53].

These events occur on each monomer unit of chitosan with a size of not more than 1 nm, which is many times smaller than the size of chitosan nanoparticles. Depending on the packing density in nanoparticles up to 50 nm in size, active centers can form, the sizes of which are much smaller than the molecule itself. Thus, one nanoparticle has several functionally significant sites of different sizes. The nanometer size of the chitosan core molecules is a fundamental factor for controlling a specific inflammatory process. Due to its nanosize, it is possible to overcome the enzymatic or adsorption barrier for successful transport through the native tissue compartment.

The degree of incorporation of hydrophobic molecules, such as cholesterol or fatty acids, into the composition of the hydrophilic polymer affects the size of self-assembled amphiphilic nanoparticles [53,54]. It is known that the hydrophobic modification of a polysaccharide, e.g., pullullan, with cholesterol leads to the formation of a water-soluble system, a nanohydrogel, which can deliver a hydrophobic compound through a tissue compartment [55,56,57]. This hydrophobic modification can be applied to chitosan both under laboratory conditions and for the transport of cholesterol and other lipid-containing macromolecules through tissues [58,59]. It is assumed that the rapid transport of modified chitosan into, for example, the subintimal zone of the vascular wall can lead to the binding to hydrophobic groups of free crystalline cholesterol or fatty acids by encapsulating them into a core of nanoparticles [60].

At the same time, these amphiphilic complexes become water-soluble [61] and acquire the ability to encapsulate lipids and penetrate through biological barriers outside the cells, for example, trans-dermally [62]. The implantation of the polymer in the perivascular zone means a local reversal of the cholesterol content. Thus, implantation of the polymer in the perivascular zone may lead to a local change in the cholesterol content. Along with this mechanism of decholesterolization, as indicated above, there is a release of cholesterol from residual cells under the intimal membrane and the formation of transmembrane flows of cholesterol and lipoproteins in the core of the sulfated chitosan nanosystem [63].

This flow outside the outer shell of the vessel can be provided by an electrostatic gradient, which is created due to the polycationic properties of chitosan [60,64]. Artificial minimally invasive dislocation of chitosan nanoparticles in the fascial sheath of the main vessels provides this gradient [65,66] and can be designed for the directed flow of the lipid mass towards the adventitia.

The results of this study show the high efficiency of lipid resorption when using highly sulfated chitosan after its implantation into the para-adventitial zone of a vascular wall affected by atherosclerosis, whereas control salt water-soluble derivatives of chitosan are less efficient. These data indicate that the proposed product is promising for local regulation of atherogenic inflammation.

## 2. Materials and Research Methods

### 2.1. Simplified Method for the Synthesis of Sulfoethylated Chitosans

A simplified method of sulfoethylation of chitosan excludes hours-long extraction in a Soxhlet apparatus. Sonat chitosan (250 kDa, 3.30 g, 0.02 mol) and sodium 2-bromoethanesulfonate (4.22 g, 0.02 mol) were thoroughly mixed in dry form in a wide tube for 3 min. Water (30 mL) was added to the dry mixture, which was heated at 70 °C for 24 h. Every three hours, the reaction mixture was thoroughly stirred. A solution of NaOH (0.53 g, 0.013 mol) in water (35 mL) was added, and the reaction mixture was cooled to 20 °C. The reaction mixture was intensively stirred for 10 h, followed by shaking for 15 h. The resulting homogeneous mixture was added to acetone (200 mL) with vigorous stirring. The precipitated sulfoethylchitosan was dried at 40–50 °C and atmospheric pressure to constant weight. The resulting product was crushed by grinding in a mortar or a ball mill. The final product (4.5 g) was obtained with a degree of sulfoethylation of 60%, which was confirmed by elemental analysis: M (C_8_H_15_NSO_7_Na) = 292, found, %: C 38.2; H 5.8; N 6.4; S 6.6; calculated, %: C 37.6; H 5.8; N 6.3; S 6.5.

We used Sonat chitosan (250 kDa), LLC NATEK chitosan (100 kDa), and sodium bromoethylenesulfonate (Acros Organics). The IR spectra of the compounds were recorded on a Nicolet iS10 spectrometer. The elemental analysis was performed on a EURO EA 3000. 1H and 13C NMR spectra were recorded on a Bruker AV-400 spectrometer at the Vorozhtsov Novosibirsk Institute of Organic Chemistry (Siberian Branch of the Russian Academy of Sciences).

### 2.2. Medical and Biological Aspects of the Study

The work was performed following ethical principles established by the European Convention for the Protection of Vertebrate Animals used for Experimental and Other Scientific Purposes (Strasbourg, 18 March 1986, adopted on 15 June 2006). All manipulations with the animals were performed following the regulations specified in the Guide for the Care and Use of Laboratory Animals (National Research Council, 2011). The work was approved by the Ethics Committee of the Voino-Yasenetsky Krasnoyarsk State Medical University.

The experimental animals (rabbits) were anesthetized by intramuscular administration of a solution of hexenal at a dose of 3–5 mg of the drug per 100 g of body weight. We used 36 male rabbits of the Shinhilla breed weighing 3.5 ± 0.5 kg. The animals were divided into six groups, with six animals in each group. The first group included intact animals on a standard vivarium diet. The second group included rabbits that were kept on a cholesterol diet for 110 days. The third group consisted of rabbits that were kept on a cholesterol diet for 80 days, followed by implantation of 4 mL of 1% water-soluble sulfated chitosan gel (O,N-(2-sulfoethyl)chitosan; sulfoethylation degree, 60%; mol weight of the structural unit, M (C_8_H_15_NSO_7_Na) = 292 g/mol) in the para-adventitial fascial sheath of the main arteries of the left hind limb in the ankle joint area. The fourth group included animals that were kept on a standard vivarium diet and subjected to the implantation of the same preparation of chitosan (4 mL) in the perivascular fascial sheath of the main arteries of the left hind limb. The fifth and sixth groups consisted of 12 rabbits that received 1% water-soluble gel of chitosan ascorbate (molecular weight, 100–700 kDa; deacetylation degree (DD), 92–95%) in the left hind limb. We used quaternized chitosan ascorbate gel as a control implant instead of the initial water-insoluble dry chitosan. The use of experimental and control implants in intact animals is required to confirm the known angiogenic effect of the presence of chitosan in tissues regardless of the presence of atherogenic inflammation (Table 1).

### 2.3. Modeling the Early Stage of Atherosclerosis

The animals were kept under vivarium conditions in AWTech Eurasia Plus cages (795 × 745 × 1910) on a cholesterol or cholesterol-free diet. The cholesterol diet for rabbits was designed for the formation of the initial stages of atherosclerosis with the formation of soft atherosclerotic plaques in the subintimal zone of the great vessels. The diet included 0.8 g of cholesterol (BioChemica) per 1 kg of rabbit body weight in unrefined sunflower oil and bread bran. Rabbits received this mixture with free access to water daily throughout the experiment for 110 days. Raw vegetables were added to the diet once a week.

On the 1st, 80th, 100th, and 110th days of the experiment, the microcirculation index was evaluated in two groups of animals at three points on the left and right hind limbs (Figure 1), and blood sampling was taken for the analysis of the lipid spectrum. The blood microcirculation parameters were studied using a laser analyzer of blood microcirculation (LACC-02, Russia). The experimental animals were withdrawn from the experiment on the 110th day. For histological studies, a complex of soft tissues was isolated in all groups of animals at the level of the middle third of the lower leg and thigh of both limbs including the main neurovascular bundle.

### 2.4. Morphological Analysis of the Vascular Wall

The tissue samples were fixed in 10% neutral buffered formalin and embedded in paraffin according to the standard technique, followed by the formation of thick histological sections (3–4 μm) using a Leica automated system ASP 300S (Leica Microsystems, Germany). The sections were stained with hematoxylin, eosin, and Sudan III. The micro-specimens were subjected to survey microscopy and morphometric evaluation using the JMicroVision v. 1.2.7 software. The measurements included the wall thickness of the femoral artery and the saphenous artery and their average lumen diameter and area. Each parameter was measured in 10 fields of view in each micro-specimen at magnifications of 100, 200, and 400. Microphotographs were obtained using an Axio Imager A1 microscope with an Zeiss AxioCam MRc5 photography system (Carl Zeiss Microspcopy GmbH, Jena, Imaging Associated Ltd, Germany) and Axio Vision software rel. 4.8 and Micromed Images with modules for automatic and manual recognition and calculation of the geometric parameters of micro-objects, Micromed Statistica software for statistical processing of measurement results (Carl Zeiss, Germany) at magnifications of 100, 200, and 400.

The following criteria were used for the morphometric evaluation of rabbit arteries: the specific volume of the artery wall (Vvaw), the specific volume of the lumen of the artery (Vval), the specific volume of the middle lining of the artery (Vvam), the coefficient of smooth myocytes in the subintimal region (calculated as the ratio of the numerical density of subintimal myocytes to the area of the middle membrane of the femoral artery), the number of vessels in the para-adventitial area excluding large main vessels, and the presence of xanthoma cells (assessed in the form of a dichotomous variable, i.e., yes–no).

### 2.5. Biopolymer Implantation Technique

The implantation of biopolymers was carried out under aseptic conditions on the 80th day of feeding with a cholesterol diet as follows. Under general anesthesia, the skin and subcutaneous fat were dissected on the left hind limb in the projection of the neurovascular bundle of the lower third of the leg. The sulfated chitosan gel (4 mL) was injected into the fascial-muscular sheath of the artery using an insulin syringe with a plastic cannula (Figure 2). The wound was closed with single-row sutures, followed by the application of an aseptic gauze bandage. The dressing was changed daily with the use of antibacterial agents until the end of the experiment. An identical manipulation was performed in the 2nd group of animals on a standard diet [66]. The results are presented in Table 1.

### 2.6. Biochemical Research Methods

In three groups of rabbits, venous blood was taken at the start of the experiment and on the 80th, 100th, and 110th days of feeding with a cholesterol or cholesterol-free diet. The blood plasma proteins and lipid spectrum (total protein level, albumin, total cholesterol, high-density lipoproteins (HDL), low-density lipoproteins (LDL), very low-density lipoproteins (VLDL), triglycerides (TG), and atherogenic coefficient) were evaluated using a Beckman Culter AU-5800 for biochemical analysis. All bench manipulations with blood plasma were carried out in compliance with the control and experimental conditions. All computational operations on the results of the biochemical analysis were reduced to the value of a 10-fold dilution of blood plasma.

A fragment of the main artery was taken from all animals on the 110th day of the experiment to obtain the morphometric characteristics of the vascular wall.

Bench experiments with normal human blood plasma consisted of first diluting it 2-fold with a 0.9% NaCl solution, followed by mixing with 0.5% gel of sulfoethylated chitosan at a ratio of 1: 1 for 0.25, 0.5, 1, 2, 3, and 4 h at 22 °C. The final mixtures were diluted 10-fold with a 0.9% NaCl solution and kept at 22 °C for 15 min, followed by centrifugation at 1000 rpm in a horizontal rotor at 10 °C for 10 min. The supernatant (1.5 mL) was used for biochemical testing.

### 2.7. Statistics

Statistical processing of the results was carried out using the SPSS20 program. Descriptive statistics are represented by absolute values and statistical coefficients. Pairwise comparison was carried out according to the Mann–Whitney test. Differences were considered statistically significant at *p* < 0.05.

## 3. Results

### 3.1. Synthesis of Sulfoethylated Chitosan

An improved method for the synthesis O,N-(2-sulfoethyl)chitosan was developed.

We found that there was no interaction between chitosan and sodium 3-chloro ethansulfonate or 2-bromoethansulfonic acid. Taking this fact into account, we can assume that the sulfoalkylation of chitosan under the conditions of gel synthesis includes preliminary dehydrohalogenation followed by Michael’s addition of sodium ethylenesulfonate to the polymer. This is why chitosan does not undergo sulfoalkylation reactions under these conditions with haloalkanesulfonic acids, which are incapable of dehydrohalogenation (Figure 1).

The structure of synthesized O,N-(2-sulfoethyl)chitosan was confirmed by physical and chemical methods.

NMR data for O,N-(2-sulfoethyl)chitosan are presented in Figure 3a,b. ^1^H NMR (400.13 MГц, D_2_O): δ 2.72 (br t, ⅓ H-2), 2.86 (br t, ⅔ H-2), 3.03–3.28 (br m, SCH_2_ and NCH_2_), 3.51–3.75 (br m, H-3, H-4, and H-5), 3.75–4.01 (br m, H-6), 4.43–4.56 (br m, H-1). ^13^C NMR (projection from ^1^H-^13^C HSQC, 100.61 MГц, D_2_O): δ 43.6 (NCH_2_), 50.2 (SCH_2_), 56.5 and 62.4 (C-2), 60.1 (C-6), 73.0, 74.0, 74.8, 78.1 (C-3, C-4 and C-5), 102.3 (C-1).

The IR spectrum of O,N-(2-sulfoethyl)chitosan (Figure 3d), in contrast to the original chitosan (Figure 3c), contains intense absorption bands at 1159, 1072, and 1043 cm^−1^, which correspond to the S=O stretching vibrations, and bands at 744 and 594 cm^−1^, which correspond to stretching vibrations of the -C-SO_3_ group.

### 3.2. Lipid Spectrum of Blood Plasma of Rabbits under the Influence of Sulfated Chitosan

Consumption of the cholesterol diet (ChD) for 80 days caused hyperlipidemia in rabbits, as evidenced by the increase in the lipid spectrum in the blood plasma of rabbits compared to intact animals. The results show that feeding rabbits with a cholesterol diet for 80 days led to an increase in total cholesterol, TG, LDL, vLDL, and the atherogenic coefficient by factors of 29, 2.5, 61, 2.4, and 7, respectively (Table 2).

The lipid spectrum of the blood plasma of rabbits on the 100th day of receiving ChD was evaluated 20 days after implantation of 1% water-soluble sulfated chitosan gel into the fascial-muscular sheath of the left femoral artery. The analysis showed that the levels of all lipid fractions were significantly higher than in rabbits without ChD but with implantation of the biopolymer. The TG level was 0.69 ± 0.14 in animals without ChD vs. 5.91 ± 1.63 in animals treated with ChD (*p* < 0.05), total cholesterol was 0.77 ± 0.14 and 30.06 ± 4.35, respectively (*p* < 0.001), HDL was 0.37 ± 0.02 and 3.00 ± 0.74, respectively (*p* < 0.005), LDL was 0.25 ± 0.02 and 24.36 ± 3.51, respectively (*p* < 0.001), vLDL was 0.31 ± 0.06 and 2.68 ± 0.74, respectively (*p* < 0.05), and AC was 0.6 ± 0.08 and 10.51 ± 2.49, respectively (*p* < 0.005). These values indicate the progression of atherogenic inflammation and the absence of any signs of correction of hyperlipidemia during attempts to reconstruct individual fragments of the great vessels of the limb.

The analysis of the blood serum lipid profile on the 110th day of the cholesterol diet showed that the levels of TG, Ch, HDL, and vLDL in rabbits gradually increased despite the implantation of 1% water-soluble sulfated chitosan gel. These indicators, in comparison with the results in the group of animals with implantation of a similar polymer, were significantly higher and were 3.69 ± 0.85 and 1.35 ± 0.17 (*p* < 0.05) for TG, 33.52 ± 1.70 and 0.65 ± 0.26 (*p* < 0.001) for Ch, 0.69 ± 0.09 and 0.39 ± 0.07 (*p* < 0.05) for HDL, and 1.68 ± 0.39 and 0.61 ± 0.08 (*p* < 0.05) for vLDL, respectively. These results indicate an increase in hyperlipidemia and the progression of atherosclerosis, as well as the absence of a clear correction of the levels of lipids in the blood plasma of rabbits when trying to reconstruct the left femoral artery using the polymer injection.

Preliminary studies on a model of experimental atherosclerosis in 42 laboratory Wistar rats [67] showed that hyperlipidemia, which was developed after a cholesterol diet for 60 days (30% vegetable oil, 2.4% cholesterol, 0.12% 6-methyl-2-thiouracil, 0.06% cholic acid, and 200,000 units of vitamin D2), was found simultaneously in both blood plasma and the wall of the femoral or external iliac arteries of the hind limb (Table 3 and Table 4). Several polymers were used for a comparative analysis of the efficiency of extracting lipid fractions from the vascular wall, i.e., 1% gel O,N-(2-sulfoethyl)chitosan (sulfoethylation degree, 60%; Mw of the structural unit, 292 g/mol); 1% gel chitosan ascorbate (Mw, 100–700 kDa; DD, 92–95%); 1% gel chitosan hydrochloride (Mw, 100 kDa; DD, 87%); 1% gel carrageenan, BCh RAS (Pacific Institute of Bioorganic Chemistry, Vladivostok); and 1% gel polyethylene glycol. Extraction of lipids from the vascular wall was carried out according to a well-known method using organic solvents [67]. The comparative analysis showed (Table 3) that the introduction of experimental and control implants in the segment of the femoral artery was accompanied by a significant decrease in the fractions of total lipids and triglycerides in the vascular wall. The drop in levels was especially noticeable when using the sulfated form of chitosan and chitosan ascorbate.

It is very important that the levels of lipid fractions in the overlying segment, the iliac artery in rats, indicated a weak effect of decholesterolization (Table 4). A significant decrease in the lipid fractions occurred only locally (in the femoral segment of the artery) in accordance with the level of polymer administration.

The positive results of the study of lipid fractions in rats allowed the authors to focus their attention on the use of chitosan ascorbate and sulfoethylated chitosan in a model of atherosclerosis in rabbits.

The TG, Ch, LHL, LDL, vLDL, and AC levels were 0.69 ± 0.14 without ChD and 5.91 ± 1.63 with ChD (*p* < 0.05), .77 ± 0.14 without ChD and 30.06 ± 4.35 with ChD (*p* < 0.001), 0.37 ± 0.02 without ChD and 3.00 ± 0.74 with ChD (*p* < 0.005), 0.25 ± 0.02 without ChD and 24.36 ± 3.51 with ChD (*p* < 0.001), 0.31 ± 0.06 without ChD and 2.68 ± 0.74 with ChD (*p* < 0.05), and 0.6 ± 0.08 without ChD and 10.51 ± 2.49 with ChD (*p* < 0.005), respectively.

The analysis of the lipid spectrum of the blood plasma of rabbits on the 100th day of ChD (20 days after implantation of sulfated chitosan gel in the para-adventitial area of the left femoral artery) showed the progression of atherogenic inflammation and the absence of any signs of correction of hyperlipidemia during attempts to reconstruct individual fragments of the main vessels of the limb (Table 5). The TG, Ch, LHL, LDL, vLDL, and AC levels were 0.69 ± 0.14 without ChD and 5.91 ± 1.63 with ChD (*p* < 0.05), 0.77 ± 0.14 without ChD and 30.06 ± 4.35 with ChD (*p* < 0.001), 0.37 ± 0.02 without ChD and 3.00 ± 0.74 with ChD (*p* < 0.005), 0.25 ± 0.02 without ChD and 24.36 ± 3.51 with ChD (*p* < 0.001), 0.31 ± 0.06 without ChD and 2.68 ± 0.74 with ChD (*p* < 0.05), and 0.6 ± 0.08 without ChD and 10.51 ± 2.49 with ChD (*p* < 0.005), respectively. The analysis of the lipid spectrum of the blood plasma on the 110th day of the cholesterol diet (30 days after implantation) showed that the levels of TG, Ch, LHL, and vLDL in rabbits gradually increased. This indicates an increase in hyperlipidemia and the progression of atherosclerosis, as well as the absence of an obvious correction of the levels of lipids in the blood plasma of rabbits during an attempt to reconstruct the left femoral artery with sulfoethylated chitosan. The analysis of the effect of implantation of 1% chitosan ascorbate gel showed a similar result (Table 5).

The results of polymer implantation in the para-adventitial sheath of the main arteries of the extremities confirm that a significant decrease in lipid fractions occurred locally following polymer administration.

### 3.3. Signs of Cholesterol Extraction in Experimental Atherosclerosis

We performed a morphological study of the arteries of the right and left hind limbs of rabbits, which were kept on a cholesterol diet or a standard vivarium diet for 110 days and then injected with sulfated chitosan gel and chitosan ascorbate in the para-adventitial space of the left hind limb. The degree of edema of the intima of the main artery, the outer and inner diameters of the vessel, the presence of xanthoma cells in its wall, the numerical density of smooth myocytes, and the degree of vascularization of the para-adventitial space were assessed.

Keeping rabbits on a cholesterol diet for 110 days (Figure 4) compared with the initial norm (Figure 5) led to a significant restructuration of the vascular wall, i.e., narrowing of the lumen of the artery, thickening of the vessel wall as a result of inflammatory edema, the formation of numerous xanthoma cells (Figure 4 and Figure 6) filled with cholesterol crystals, and myocyte disorientation (Figure 7).

In the subintimal space in the great vessels of the right lower limb, soft atherosclerotic plaques were formed, thus disrupting the spherical shape of the vessel lumen. Fibroblasts and migrating smooth muscle cells actively proliferated in the subintima (Figure 8).

Thirty days after the injection of 1% sulfated chitosan into the para-adventitial area of the main artery of the left lower limb, there was a noticeable decrease in arterial wall edema in the presence of an active productive inflammatory reaction in the form of numerous macrophages and giant multinucleated cells (Figure 9). At the same time, a positive trend of a reduction in cells filled with cholesterol was observed. The proliferation of myocytes in the middle layer of the vessel significantly decreased, and a zone rich in microvessels was formed at the site of the implantation of sulfated chitosan (Figure 10). The presence of a large number of macrophages and the formation of a large number of giant multinucleated cells indicate active polymer degradation in the perivascular space (Figure 11 and Figure 12). One month after the implantation of the chitosan gel into the fascial vascular sheath of the main neurovascular bundle of the left hind limb of the rabbit, we observed signs of decholesterolization of the artery wall. These signs were the formation of active neo-angiogenesis, restoration of the normal structure of the layers, an increase in the average size of the arterial lumen, the detection of a small number of xanthoma cells in the subintimal zone, and low proliferation of smooth muscle cells in the middle layer. At the same time, the complete degradation of the polymer was not yet completed (Figure 13 and Figure 14).

### 3.4. Morphometric Characteristics of the Wall of Main Arteries of Rabbit Hind Limbs after Injection of Sulfated Chitosan into the Para-Adventitial Area

The results in Table 6 show that the specific volume of the vascular wall of the artery of the lower leg of intact animals was 39% lower than that of rabbits that received a cholesterol diet for 110 days (50.17 ± 5.80% vs. 82.04 ± 8, respectively, *p* < 0.05). This morphometric indicator significantly decreased by 27% and 15.2%, respectively, when implanting 1% water-soluble sulfated chitosan gel into the para-adventitial zone of the left leg of rabbits (60.12 ± 3.58%) compared to the right limb (82.46 ± 7.65%, no biopolymer introduced, *p* < 0.05). Under ChD and 30 days after implantation of control chitosan ascorbate, the specific volume of the artery wall of the right limb decreased by only 14% compared with the right limb of rabbits and by 26% after implantation of 1% sulfoethylchitosan gel. The implantation of the biopolymer into the perivascular space of the tibia in rabbits kept on a standard vivarium diet did not lead to a significant decrease in the specific volume of the vascular wall of the tibial artery. However, it should be noted that there was a clear trend toward a decrease in this indicator after implantation compared to the situation when the biopolymer was not injected.

Consumption of a cholesterol diet for 110 days led to a significant decrease of 40% in the specific volume of the lumen of the artery of the right leg (16.47 ± 2.48%) compared with intact animals (27.69 ± 4.65%) (*p* < 0.05). The comparison of this morphometric parameter for the left and right lower legs of experimental animals indicates that the implantation of 1% water-soluble sulfated chitosan gel leads to an increase of 87% (30.87 ± 3.4%) in this parameter in the left lower leg. The implantation of chitosan ascorbate gel did not affect this parameter. The introduction of a chitosan biopolymer into the para-adventitial space of the shin in rabbits kept on a standard vivarium diet also insignificantly affected the specific lumen volume of the artery of the left shin. It is important to emphasize that the implantation of both experimental and control polymers in the intact left limb did not increase the specific lumen volume of the artery compared to that in the right leg.

The analysis of the specific volume of the media of the artery of the lower leg shows that this parameter in intact animals was 34% lower than that in the lower leg of the right limb of rabbits that received a cholesterol diet for 110 days (50.86 ± 5.68% vs. 76.82 ± 4.71%, *p* < 0.05, respectively). When using a cholesterol diet, this morphometric criterion significantly decreased by 22% after implantation of 1% sulfoethylated water-soluble chitosan gel (60.13 ± 3.22%) in the para-adventitial zone of the left leg of rabbits (60.13 ± 3.22%) compared to the right limb (76.82 ± 4.71%, *p* < 0.05). The introduction of 1% chitosan ascorbate gel into the fascial arterial sheath did not change the indicated characteristic of the artery.

The analysis of subintimal myocytes of the artery of the right leg (Table 6) shows that consuming a cholesterol diet for 110 days led to the significant migration (64%) of these cells from the middle layer of the vessel (the content of these cells was 85.56 ± 9.34 vs. 52.09 ± 7.04, *p* < 0.05, in intact animals). The comparison of this morphometric index in the left and right lower legs of experimental animals indicates that implantation of 1% sulfated chitosan gel led to a decrease of 27% in this parameter in the left lower leg (the area of the biopolymer layer was 62.76 ± 4.71 vs. 85.56 ± 9.34, *p* < 0.05, in the right leg). The implantation of sulfoethylated chitosan into the para-adventitial zone of the shin in rabbits kept on a standard vivarium diet insignificantly affected the coefficient of subintimal myocytes of the artery of the left shin. The general range of dispersion of the average values of the studied parameter was 50.39–53.98, and the range of differences in the right and left limbs was 0.5–3.59. The use of the control polymer showed a negligible effect on the correction of the imbalance of subintimal myocytes.

On the 110th day of the cholesterol diet (ChD), after the implantation of 1% sulfoethylated chitosan gel, we observed a significant increase in the number of vessels in the para-adventitial space of the leg (direct dislocation of the biopolymer) compared to the limb with no injection. The dislocation of chitosan in the fascial bed of the main artery for 30 days led to a 56% increase in the number of vessels in the para-adventitial space at the site of polymer degradation. In the left and right limbs, these parameters were 28.75 ± 3.11 and 18.42 ± 3.92 (*p* < 0.05), respectively. Moreover, the implantation of the polymer into the left limb in intact animals led to a significant increase of 88% on the 30th day in the number of vessels in the resorption zone of the chitosan construct compared to the right limb (23.3 ± 3.45 vs. 12.38 ± 2, *p* < 0.05, respectively). The use of 1% chitosan ascorbate gel in the control series showed a similar result; i.e., the introduction of water-soluble chitosan into the para-adventitial zone of the artery led to the significant formation of new capillaries.

The study of the vascular wall of the femoral artery of intact animals in comparison with the vessels of the animals on a cholesterol diet showed that the changes in these criteria were similar to those for the vessels of the leg. The specific volume of the vascular wall of the femoral artery in intact animals was significantly lower than that in rabbits on a cholesterol diet (56.64 ± 3.49% vs. 75.08 ± 4.03%, *p* < 0.05, respectively). This morphometric indicator significantly decreased by 31% only after the implantation of 1% sulfoethylated chitosan gel in the para-adventitial space of the left limb of the ChD-treated animals (51.75 ± 3.75% vs. 75.08 ± 4.03% in the right limb without biopolymer, *p* < 0.05) (Table 7). We did not observe a significant change in the specific volume of the vascular wall of the femoral artery during implantation of the biopolymer in the para-adventitial area of the animals on the standard diet. The patterns of preferential correction in the study of the leg arteries were replicated when using the sulfated form of chitosan, in contrast to the control polymer. The characteristics of the femoral artery in the left limb in the experimental series Vvwa (%), Vvla (%), and Vvma (%) and subintimal myocyte ratio were significantly different compared to the control series and corresponded to normal numerical characteristics (Table 7).

Consuming the cholesterol diet for 110 days led to a 56% decrease in the specific volume of the right femoral artery lumen (13.07 ± 4.62%) compared to that in intact animals (29.83 ± 3.62%, *p* < 0.05). Comparison of this criterion between the left and right thigh demonstrated that the injection of sulfated chitosan led to a 90% (24.98 ± 4.04%) increase in the lumen in the femoral artery of the left limb compared to the right limb (13.07 ± 4.62%, *p* < 0.05). The injection of the biopolymer into the para-adventitial area of the vessels of the lower leg and thigh of the left limb in rabbits kept on a standard vivarium diet did not significantly affect the specific lumen volume of the artery. The overall range of dispersion of the mean values of the studied parameter was 25.69–30.52 (Table 7). A comparative study of the specific volume of the middle layer of the femoral artery of intact animals and animals on a cholesterol diet showed that a cholesterol diet for 110 days increased the thickness of this layer by 18% in the right femoral artery (59.32 ± 2.39% vs. 72.27 ± 2.14%, *p* < 0.05). This morphometric parameter significantly decreased in the femoral artery of the left limb by 11% when sulfated chitosan was implanted into the para-adventitial zone (64.13 ± 3.22% in the left limb vs. 72.27 ± 2.14% in the right limb, *p* < 0.05). The injection of the biopolymer in rabbits kept on a standard vivarium diet did not significantly affect the specific volume of the middle layer of arteries.

The proliferation of smooth muscle cells in ChD-treated rabbits indicated that their growth in the middle layer of the artery increased by 36% (*p* < 0.05). The coefficient of subintimal myocytes of the artery of the right thigh was 82.71 ± 5.69, which exceeded the value obtained for intact animals (60.62 ± 6.80, *p* < 0.05). Comparative analysis of this parameter between the arteries of the left and right thighs revealed that the injection of chitosan on the left decreased proliferation by 22.6% (64.01 ± 6.41) in comparison with the injection on the right (82.71 ± 5.69, *p* < 0.05). The introduction of chitosan biopolymers into the para-adventitial area of the shin of rabbits kept on a standard vivarium diet did not significantly affect the coefficient of subintimal myocytes of the left femoral artery.

The analysis of the number of micro-vessels in the para-adventitial zone of the thigh of the right and left hind limbs of ChD-treated rabbits showed that implantation of sulfated chitosan in the lower leg area caused a slight increase in this parameter in the overlying segment of the vessel of the left hind limb. The difference in the number of vessels in the para-adventitial space of the thigh of the right and left hind limbs of rabbits kept on a standard vivarium diet and treated with the biopolymer in the shin area was statistically insignificant.

### 3.5. Sorption of Human Blood Plasma on Sulfated Chitosan

A convincing demonstration of the restoration of important characteristics of the arterial wall using O,N-(2-sulfoethyl)chitosan allowed us to use this synthesized chitosan derivative in an in vitro experiment to evaluate its affinity for lipid fractions of blood plasma. The mixing of 10-fold diluted blood plasma with 1% gel of water-soluble sulfated chitosan in a ratio of 1: 1 (*v*/*v*) for 15 min–4 h at 22 °C led to the capture of the binding sites of lipid fractions (Table 8).

The addition of sulfoethylated chitosan to blood plasma did not affect the level of total protein. The polymer actively bound to albumin (22.1%) 15 min after sorption, and the product reduced the total cholesterol level by 6.2%, did not affect the levels of triglycerides, HDL cholesterol, or vLDL, steadily reduced the LDL cholesterol level by 12.5–22% over time, and significantly reduced the atherogenic coefficient in the process of sorption by 20–34.7%. Thus, in bench experiments, sulfoethylated chitosan was demonstrated to be a promising water-soluble polymer for a minimally invasive technology for the extraction of cholesterol/lipid fractions from biological tissues in both bench experiments and animal models. The product reduces the total cholesterol level, does not bind to HDL cholesterol, and steadily reduces LDL levels and the atherogenic coefficient.

### 3.6. Microcirculation Index (MI) Evaluated by Laser Doppler Flowmetry of the Hind Limbs of Rabbits

We found no statistically significant differences in the MI values of the right and left hind limbs in the rabbits before the implantation of biopolymers. Therefore, the numerical values were combined and averaged. The data in Table 9 show that consumption of the cholesterol diet for 80 days led to a decrease in the microcirculation index at all of the studied points compared to that in the animals on the standard vivarium diet. At point 1, which is located in the region of the inguinal fold, the perfusion unit (PU) in ChD-treated rabbits was 2.79 ± 0.33 vs. 5.32 ± 0.56 (*p* < 0.05) in intact rabbits. At point 2, where the gel implant was located just below the knee joint, the MI value was 3.43 ± 0.35 PU in ChD-treated animals vs. 7.88 ± 0.78 PU (*p* < 0.05) in rabbits without ChD. Consequently, consumption of ChD for 80 days led to significant decreases of 48% and 56% in tissue perfusion at points 1 and 2, respectively.

The MI value at the second registration point increased by 80% 20 days after the implantation of sulfated chitosan into the perivascular zone in the region of the left leg compared to the initial MI level in both rabbits treated with ChD and those kept on a standard vivarium diet (Table 9).

When comparing the values of tissue perfusion in the same segments of opposite limbs, it is important to note that this parameter in the left segment increased by 86% (6.16 ± 0.51 PU vs. 3.43 ± 0.35 and 3.31 ± 0.53 PU, respectively, *p* < 0.05).

The MI value decreased in both the left and right hind limbs 30 days after the implantation of the biopolymer, although this parameter was 34% higher in the left limb (4.36 ± 0.25 PU vs. 3.25 ± 0.29 PU, *p* < 0.05, respectively). When the polymer was implanted in intact animals, the MI value at point 2 of the left hind limb increased by 26% (9.70 ± 0.27 PU vs. 7.67 ± 0.34 PU, *p* < 0.001) and by 31% (9.87 ± 0.45 PU vs. 7.52 ± 0.61 PU, *p* < 0.05) on the 20th and 30th days after surgical intervention, respectively. A comparative analysis of the perfusion index values after injection of 1% chitosan ascorbate gel into the left limb showed that there was a slight tendency to increase the level of soft tissue blood supply in the area of point 2 within 30 days after the polymer implantation.

Thus, the introduction of a 1% gel of water-soluble sulfated chitosan into the perivascular fascial space of the great vessels leads to the significant extraction of low-density lipid fractions from the blood plasma and vascular wall of the experimental left hind limb compared to the right limb of rabbits on a long-term cholesterol diet. In addition, the implantation of sulfated chitosan rebuilds the quantitative characteristics of the vessel layers towards a normal structure and increases the degree of perfusion of the limb tissues.

## 4. Discussion

It is known that the formation of early plaques in the subintimal region of the artery is characterized by the spread of the inflammatory reaction beyond the adventitia [23]. We proposed implanting the sulfoethylated form of highly deacetylated chitosan directly into the problem area of the main artery, i.e., into the para-adventitial zone over a large area affected by early soft atherosclerotic plaques. The work was performed on a rabbit model. The technology is based on the ability of a sulfated biopolymer to act as a delivery system to the zone of damage and its well-known high penetrating ability through the intercellular space of the tissue. The polymeric construction of chitosan is biodegradable and biocompatible, has low toxicity, and exhibits a high affinity for cholesterol and low-density lipoproteins. The proposed simplified technology for chemical synthesis produces chitosan with a high sulfation degree. The dislocation of sulfated chitosan near cell membranes in the subintimal region suggests its active interaction with cholesterol in macrophages, its binding to extracellular and intracellular cholesterol, and transport of its water-soluble electrolyte complex with lipid mass [61] into the para-adventitial region in the composition of chitosan nuclei [60]. It is assumed that the targeted transport of lipid fractions occurs due to an electrostatic gradient [64,65,66], which is caused by artificial minimally invasive dislocation of chitosan [47] in the fascial sheath of large vessels. Our study convincingly shows the local release of cholesterol and low-density lipoproteins from the arterial wall. The high LDL cholesterol content in both the blood plasma and the wall of the control vessel in the model of early atherosclerosis was associated with its low content in the experimental limb of the same animal. Evaluation of lipid fractions in the adjacent segment of the artery showed that decholesterolization occurred only in the segment that contained the implanted polymer. The decrease in atherogenic inflammation was accompanied by the restoration of important characteristics of the artery, such as the specific volumes of the wall, artery lumen, and artery media, the coefficient of subintimal myocytes, and the ratio of the numerical density of subintimal myocytes to the area of the middle layer of the artery.

The goal of the study was to evaluate the effect of resorption of soft atherosclerotic plaques in the subintimal region of the main arteries of the hind limbs in a rabbit model of atherosclerosis and demonstrate a reduction in the signs of atherogenic inflammation in the vessel wall. The comparative analysis of fragments of the iliac, femoral, and tibial arteries at similar points of the right and left hind limbs made it possible to identify significant local morphological differences in the same animal treated with a long-term cholesterol diet. Para-adventitial implantation of water-soluble chitosan into one of the limbs created a one-sided effect of morphological reconstruction. Consumption of the proposed cholesterol diet for four months in both rats and rabbits led to very high levels of low-density and very low-density lipoproteins in the blood plasma and artery wall. The decholesterolization effect in the experimental fragment of the artery against the background of high hyperlipidemia indicates the important role of chitosan injected in the para-adventitial zone.

It is known that atherogenic inflammation creates an angiogenic effect in the vascular wall, which can increase the influx of cholesterol fractions into the subintimal region, accelerate the formation of atherogenic plaques, and reduce tissue perfusion. At the same time, the presence of chitosan in tissues is also known to promote an angiogenic effect, which, on the contrary, improves their perfusion characteristics. Implantation of water-soluble sulfoethylated chitosan in a limited area of the arterial wall leads to a high angiogenic effect that suppresses the negative effect of hyperlipidemia. It should be noted that the absence of hyperlipidemia does not negate the high angiogenic effect of chitosan in the experimental limb, which leads to an increase in the level of tissue perfusion.

Preliminary studies on the model of atherosclerosis in rats have shown that all chitosan derivatives can decrease the levels of lipid fractions in the artery wall of the experimental hind limb to varying degrees. The highest results were demonstrated for the chitosan ascorbate gel and sulfated form of chitosan. To confirm the local antiatherogenic effect, these two products were used as reference samples when studying a larger model of atherosclerosis in rabbits.

Implantation of water-soluble sulfoethylated chitosan into the para-adventitial zone for 30 days showed a prolonged effect of the polymer and its advantages over 1% chitosan ascorbate gel, which allows one to recommend the former product for further research. These advantages are a high degree of restoration of the artery lumens, a decrease in the degree of edema of the vascular wall, elimination of the disintegration of the layer of subintimal myocytes, and a sharp decrease in the number of cholesterol-containing cells in the subintimal zone. Implantation of the control and experimental polymers into the left limbs of rabbits that were on the high-cholesterol diet confirmed a high angiogenic effect in this limb compared to the level of tissue perfusion in the right limb. This result was also characteristic of intact animals but only when using the sulfoethylated form of chitosan.

Thus, the technology for minimally invasive implantation of highly sulfated chitosan into the para-adventitial zone of the main artery affected by atherogenic inflammation is a promising approach to the local treatment of the atherosclerotic process at any location.

## 5. Conclusions

The results of the study confirm the preliminary conclusions about the possibility of the regulation of cholesterol metabolism in the wall of the main vessels in experimental atherogenic inflammation and the formation of soft plaques in the subintimal region after the implantation of sulfated chitosan directly into the lesion. It is assumed that highly sulfated chitosan enables the formation of a transport system capable of capturing large hydrophobic compounds, e.g., cholesterol, in the vessel wall of laboratory animals with atherosclerosis. The technology for the chemical synthesis of quaternized sulfated chitosan makes it possible to obtain a water-soluble gel (O,N-(2-sulfoethyl)chitosan; sulfoethylation degree, 60%; mol. weight of the structural unit C_8_H_15_NSO_7_Na, 292 g/mol), which can be used for implantation in the para-adventitial zone of the vascular system to provide clear signs of decholesterolization of the vessel wall. The contact of human blood plasma with sulfated chitosan demonstrates its high binding activity to low-density lipoproteins and very low-density lipoproteins, thus decreasing the plasma atherogenic coefficient. Long-term exposure to sulfated chitosan hydrogel on a large section of the main artery promotes an active angiogenic reaction and increases the perfusion capacity of the vessels. Extensive testing and certification of the stable technology for the synthesis of water-soluble quaternized sulfated chitosan will make it possible to recommend this product as an implant not only for the treatment of chronic ischemia of the lower extremities but also for the restoration of regular characteristics of vessels at any location. Thus, the technology of minimally invasive implantation of highly sulfated chitosan into the para-adventitial zone of the main artery affected by atherogenic inflammation is promising for the local treatment of the atherosclerotic process at any location.

## Data Availability

All data generated or analyzed in this study are included in the published article.

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
