# Peer review of "Synthesis, Chemical and Biomedical Aspects of the Use of Sulfated Chitosan"

_polymers, 2022, doi:10.3390/polym14163431_

Round 1
Reviewer 1 Report
I think the current revised version seems OK for me
Author Response
Response to Reviewer 1 Comments
Point 1: Moderate English changes required
Response 1:
Moderate changes were made to the text. A text including the results of their own experiments was added to page 23.

Reviewer 2 Report
The paper entitled "Synthesis, chemical and biomedical aspects of the use of sulfated chitosan" by Igor Nikolaevich Bolshakov and co. describes the synthesis of sulfated chitosan and its characterisation, in an animal model of early atherosclerosis
The paper is interesting and well written, but some minor revisions must be done, as follows:
Page 23 - the Discussion part is mainly based on bibliographic content, please introduce the discussion based on experimental results, especially the figures and tables obtained in experimental results
Author Response
Point 1: Page 23 - the Discussion part is mainly based on bibliographic content, please introduce the discussion based on experimental results, especially the figures and tables obtained in experimental results.
Response 1:
Presented a discussion based on their own experimental results.
Added text:
“The goal of the study was to obtain the effect of resorption of soft atherosclerotic plaques in the subintimal region of the main arteries of the hind limbs in rabbits in the model of atherosclerosis and demonstrate the reduction of the signs of atherogenic inflammation in the vessel wall. The comparative analysis of fragments of the iliac, femoral, and tibial arteries at similar points of the right and left hind limbs made it possible to identify local significant morphological differences in the same animal treated with a long-term cholesterol diet. Para-adventitial implantation of water-soluble chitosan into one of the limbs created a one-sided effect of morphological reconstruction. The proposed cholesterol diet for four months in both rats and rabbits led to a very high level of low-density and very low-density lipoproteins in the blood plasma and artery wall. The decholesterolization effect in the experimental fragment of the artery against the background of high hyperlipidemia indicates the important role of chitosan injected in the para-adventitial zone.
It is known that atherogenic inflammation creates an angiogenic effect in the vascular wall, which can increase the influx of cholesterol fractions into the subintimal region, accelerate the formation of atherogenic plaques, and reduce tissue perfusion. At the same time, the presence of chitosan in tissues is also known to promote an angiogenic effect, which, on the contrary, improves their perfusion characteristics. Implantation of water-soluble sulfoethylated chitosan in a limited area of the arterial wall leads to a high angiogenic effect that suppresses the negative effect of hyperlipidemia. It should be noted that the absence of hyperlipidemia does not cancel the high angiogenic effect of chitosan in the experimental limb, which leads to an increase in the level of tissue perfusion.
Preliminary studies on the model of atherosclerosis in rats have shown that all chitosan derivatives can decrease the levels of lipid fractions in the artery wall of the experimental hind limb to varying degrees. The highest results were demonstrated for the chitosan ascorbate gel and sulfated form of chitosan. To confirm the local antiatherogenic effect, these two products were used as reference samples when studying a brighter model of atherosclerosis in rabbits.
Implantation of water-soluble sulfoethylated chitosan into the para-adventitial zone for 30 days showed a prolonged effect of the polymer and its advantages over 1% chitosan ascorbate gel, which allows one to recommend the former product for further research. These advantages are a high degree of restoration of the artery lumens, a decrease in the degree of edema of the vascular wall, elimination of the disintegration of the layer of subintimal myocytes, and a sharp decrease in the number of cholesterol-containing cells in the subintimal zone. Implantation of the control and experimental polymers into the left limb of rabbits, which were on the intensive cholesterol diet, confirmed a high angiogenic effect in this limb compared to the level of tissue perfusion in the right limb. This result was also characteristic of intact animals but only when using the sulfoethylated form of chitosan.”

This manuscript is a resubmission of an earlier submission. The following is a list of the peer review reports and author responses from that submission.
Round 1
Reviewer 1 Report
The manuscript describes the synthesis of sulfoethyl chitosan (SEC) and its use against atherosclerosis.
Unfortunately, even if the data might be interesting, the work has non-recoverable flaws. First, the rationale that prompted researchers to study the sulfate derivative of chitosan is not clear. The authors explain in the introduction that SEC could form disulfide bridges with glycoproteins, too bad the sulfate cannot form disulfide bridges.
The most serious defect is not having foreseen a group of animals treated with the parent chitosan to demonstrate the usefulness of derivatization.
Furthermore, the manuscript lacks the clarity of the results presentation. Here are some examples:
1) I do not understand the need to predict a group of healthy animals treated with the polymer (group 2 table 1)
2) in the methods it is claimed to have synthesized the SEC from 3 different chitosans, however only one SEC was tested
3) the synthesis schemes on page 8 are real figures that should be numbered and their content described in the caption. Besides, I don't understand the usefulness of the second scheme
4) figure 3: the IR spectrum of the SEC should be compared with that of native chitosan
5) on page 9 the authors refer to the NMR spectrum without showing it and without explaining how the degree of substitution of the sulphate groups on the polymer was determined
6) Tables 3 and 4 have the same caption but different data also with respect to those reported in the text. This is why I did not understand what they refer to. Furthermore, the groups of animals are different from those declared in the methods and reported in table 1.
Reviewer 2 Report
The abundance of tables makes it difficult to understand the main idea of the article. The title of article also does not seem to be very successful, it is necessary to specify what exactly was studied. This title is suitable for a review article. In general, the article needs to be revised, the histological drawings coincide with those of the previous article, where the co-author is one of the co-authors of this article: http://www.stm-journal.ru/en/numbers/2017/4/1395/html. These figures cannot be reused without mentioning of earlier publication. In general, data are presented that have already been published with extension.
Reviewer 3 Report
In this manuscript, the researchers developed the chemical synthesis of sulfated chitosan and its experimental verification in an animal model of early atherosclerosis. The reliable technology for the synthesis of water-soluble quaternized sulfated chitosan and its extensive testing and certification make it possible to recommend this product for the treatment of not only chronic ischemia of the lower extremities but also the restoration of vessels of any localization. However, there are some errors in this manuscript and the available data do not allow for a final conclusion. Therefore, I think this paper still major revision before acceptance:
1. Section 1.1 and section 3.1 have duplicate content on the synthesis of sulfated chitosan. Also, there should be a synthesis schematic diagram and chemical structure of the sulfated chitosan in section 1.1.
2. The section of “Introduction” needs to be refined. In section 1.2, the formation mechanism of vascular atherosclerosis and signs of cholesterol extraction is unclear. There should be a schematic diagram of the mechanism.
3. Some pictures in this manuscript are not clear enough,such as Figure 1, Figure 2, and Figure 3, which is unprofessional. More, the format of tables and figures in this manuscript needs to be beautified and uniform.
4. In section 3.1, the chemical structure and reaction process of sulfated chitosan show without legends. And, there is no 1Н NMR spectra of sulfated chitosan recorded in D2O, which is not able to allow for a final conclusion.
5. There is a problem with the numbering in Figure 4 and Figure 5, which is easy to confuse.
6. The format of references needs to be uniform, e.g., Ref.1 and Ref.2.
7. The authors could add the following references which would again increase the interest to general functional chitosan material readers: Journal of Bioresources and Bioproducts, 2021, 6(3): 223-242; International Journal of Biological Macromolecules, 2021, 185, 832-848.